

# Establishment and quality evaluation of a glioma biobank in Beijing Tiantan Hospital

Fanhong Kong[1,*], Wenli Zhang[1,*], Lin Qiao[2], Qi Li[2], Haowen Li[2], Jingli Cao[2], Wenyan He[2], Chengya Dong[2], Yanjiao He[3], Lu He[1], Li Liu[2], Weilun Fu[1], Lijun Liu[4], Zirui Li[2] and Yajie Wang[5]

[1] Beijing Tiantan Hospital, Capital Medical University, Beijing, China
[2] Clinical Medical Research Laboratory, Beijing Tiantan Hospital, Capital Medical University, Beijing, China
[3] Neuropathological department, Beijing Neurosugical Institute, Capital Medical University, Beijing, China
[4] Department of Cancer Biology, Dana-Farber Cancer Institute, Boston, MA, USA
[5] Clinical Laboratory, Beijing Ditan Hospital, Capital Medical University, Beijing, China
[*] These authors contributed equally to this work.

## ABSTRACT

**Background.** We established a glioma biobank at Beijing Tiantan Hospital in November, 2010. Specialized residents have been trained to collect, store and manage the biobank in accordance with standard operating procedures.
**Methods.** One hundred samples were selected to evaluate the quality of glioma samples stored in the liquid nitrogen tank during different periods (from 2011 to 2015) by morphological examination, RNA integrity determination, DNA integrity determination and housekeeping gene expression determination.
**Results.** The majority of samples (95%) had high RNA quality for further analysis with RIN $\geq$6. Quality of DNA of all samples were stable without significant degradation.
**Conclusion.** Storage conditions of our biobank are suitable for long-term (at least five years) sample preservation with high molecular quality.

## INTRODUCTION

Gliomas refer to a group of tumors that arise from the glial tissue of the central nervous system. Basically, they can occur anywhere in the central nervous system, both brain and spine. Gliomas are the most commonly occurring tumors of the central nervous system which make up about 27% of all brain tumors and 80% of all malignant primary brain tumors (*Agnihotri et al., 2013*; *Wilson, Karajannis & Harter, 2014*; *Aldape et al., 2015*). The incidence rate of glioma is approximately 4.67–5.73 per 100,000 persons (*Ostrom et al., 2014*). Glioblastoma (GBM) is the most malignant type of gliomas. The median survival time of patients with GBM is 12–15 months (*Bryukhovetskiy et al., 2016*).

Tumor biobanks aim to systematically collect, store, manage and utilize tumor tissues from surgical removal, blood samples and also the corresponding participant data for further use of scientific projects and more specifically, for clinical and translational

Corresponding author
Yajie Wang,
wangyajie@ccmu.edu.cn

research (*Watson, Kay & Smith, 2010*; *Kang et al., 2013*). Relying on the neurosurgery discipline, a well-managed and normative glioma biobank was established in our hospital in November, 2010. Using morphological examination, RNA integrity determination, DNA integrity determination and housekeeping gene expression determination, the quality of the glioma tissues stored in the liquid nitrogen tank for 1–6 years (collected from 2011 to 2015) was evaluated in order to provide high quality samples for medical research.

## MATERIALS AND METHODS

### Laboratory equipment

The lab is equipped with $-80\,°C$ freezers, $-150\,°C$ freezers and liquid nitrogen tanks. Each freezer has a specialized temperature monitoring system to detect real-time temperature. When the temperature inside freezers is abnormal, alarm messages will be sent to biobank managers. Central air-conditioning system keeps the room temperature around $20\,°C$.

### Collection and storage of tumor tissues

Excised glioma tissues from patients were collected immediately by trained and experienced staffs as soon as tumor samples were removed from the body during neurosurgery. Consent to use resected tissues was obtained from all cancer patients prior to surgery. This study was approved by the Ethics Committee of Beijing Tiantan Hospital (KY2014-021-02). Fresh human glioma tissues were obtained directly after surgery was completed. Blood and necrotic tissue on the surface of the samples were washed off with pre-cooling PBS. Then the tumor samples were cut into small pieces (about $0.5\,cm \times 0.5\,cm$, thickness $<0.5\,cm$) under aseptic conditions, repackaged in cryotubes and quickly frozen in liquid nitrogen tank in the operating room for temporary storage. All samples were frozen within 30 min after resection (*Tonje et al., 2016*). On the same day, they were transferred to liquid nitrogen tanks in the biobank for long-term preservation. Registration of samples were done at the same time. Clinical information of patients including name, gender, age, pathological diagnosis, clinical diagnosis and treatment were all recorded and stored in the database.

### Morphology characteristics

One hundred samples were selected to conduct the following experiments from 2011 to 2015. For each year, twenty samples were selected randomly. Firstly, the tissue fragments were fixed in 10% formalin before thawed for 24 h then embedded in paraffin. Next, five-micrometer tissue sections were cut, dewaxed and stained with Hematoxylin–Eosin (HE) in accordance with standard procedures (*Iigen et al., 2014*).The slides were observed by an experienced pathologist. Tumor samples which were comprised of $\geq 80\%$ tumor nuclei and $\leq 20\%$ necrosis ($\leq 50\%$ necrosis for GBM) were considered fit for other research.

### RNA isolation

Total RNA was isolated using TRIZOL reagent (Invitrogen, Stockholm, Sweden). A total of 1 ml TRIZOL reagent was added into an RNAse-free tube. 100 mg frozen tissue of each sample was then removed into the tube and homogenized using RNAse-free pestles. The homogenized tissue was incubated in TRIZOL reagent for 10 min at room temperature.

After those, 200 ul chloroform was added into each tube, and the tubes were manually shaken for 15 s before placed at room temperature for 10 min. The tubes were centrifuged at 12,000 rpm for 15 min at 4 °C, after which the aqueous phase was transferred to a new RNAse-free tube. Afterwards, 500 ul isopropyl alcohol was added into each tube and incubated at −20 °C for 20 min. The tubes were centrifuged at 12,000 rpm for 10 min at 4 °C. The supernatant was discarded, and 1ml 75% ethanol was added into each tube to wash RNA pellet. Finally, the tubes were centrifuged at 7,500 g for 10 min. The supernatant was discarded, and the RNA pellet was air-dried and subsequently dissolved into 30 ul RNase-free water.

## RNA yield and integrity determination

A Nanodrop spectrophotometer (Thermo Fisher Scientific, Waltham, MA, USA) was used to determine concentrations (ng/ul) and purity of RNA samples. The A260/A280 ratio was measured to indicate RNA purity. The ratio of samples with high purity is 1.8∼2.1 (*Sanabria et al., 2014*). An Agilent 2100 bioanalyzer in conjunction with the RNA 6000 Nano and the RNA 6000 Pico LabChip kits (Agilent Biotechnologies, Palo Alto, CA, USA) was used to evaluate RNA integrity, from which RNA integrity number (RIN) was calculated. A scoring system between 1 and 10 were used in the RIN software, with 1 representing degraded RNA and 10 representing very high-quality, intact RNA (*Griffin et al., 2012*). In literature, samples were divided into four quality groups according to RIN: RIN<5, not reliable for demanding downstream analysis; $5 \leq RIN < 6$, suitable for quantitative reverse transcription-PCR (RT-qPCR); $6 \leq RIN < 8$, suitable for gene array analysis; and RIN > 8, suitable for all downstream techniques (*Kap et al., 2014*). The assays were performed according to the manufacturer's instructions.

## cDNA synthesis and real-time quantitative PCR (RT-qPCR) for housekeeping genes

Extracted RNA (1 ug) was reverse-transcribed to first-strand cDNA using the PrimeScript RT reagent Kit With gDNA Eraser (TaKaRa, Shiga, Japan). Afterwards, 2ul cDNA solution was used for a 40-cycle SYBR Green PCR assay with the SYBR Premix EX Taq reagent (TaKaRa, Shiga, Japan). The same sample was run three different times in the same experiment to remove any outliers. Primer sets were as follows: human ACTB: forward primer 5′-TTAGTTGCGTTACACCCTTTCTTG -3′; reverse primer 5′- GTCACCTTCACCGTTCCAGTTTT-3′; human GAPDH: forward primer 5′-CTATAAATTGAGCCCGCAGCC-3′; reverse primer 5′-GCGCCCAATACGACCAAATC-3′, The Thermal Cycler DiceReal Time System (TaKaRa, Shiga, Japan) was used for qRT-PCR under the following conditions: 95 °C for 30 s, 95 °C for 5 s, 58 °C for 30 s 72 °C for 30 s.

## DNA isolation

100 mg frozen tissue of each sample was pulverized to isolate genome DNA. DNA isolation was accomplished using the EasyPure Genomic DNA kit (Transgen Biotech, Beijing, China) according to the manufacturer's instructions. In the end, the DNA pallet was dissolved into 50 ul ddwater (PH > 8.0).

**Table 1  Primers used for β-globin gene amplification by PCR.**

| Amplicon | Forward primer | Reverse primer | Size (bp) |
|---|---|---|---|
| I | GAAGAGCCAAGGACAGGTAC | CAACTTCATCCACGTTCACC | 268 |
| II | GCTCACTCAGTGTGGCAAAG | GGTTGGCCAATCTACTCCCAGG | 536 |
| III | ATTTTCCCACCCTTAGGCTG | TGGTAGCTGGATTGTAGCTG | 989 |
| IV | GGTTGGCCAATCTACTCCCAGG | TGGTAGCTGGATTGTAGCTG | 1,327 |

## DNA yield and PCR for housekeeping genes

A Nanodrop spectrophotometer (Thermo Fisher Scientific, Waltham, MA, USA) was used to determine concentrations (ng/ul) and the A260/A280 of DNA samples. After that, 50ng genomic DNA was used for a 40-cycle SYBR Green PCR assay with the SYBR Premix EX TaqII reagent (TaKaRa, Shiga, Japan). Each measurement was performed in triplicate to remove any outliers. Primer sets were as follows: human ACTB: forward primer 5′-AAGACCTGTACGCCAACACA-3′; reverse primer 5′-CTGGATGTGACAGCTCCCC-3′; the primer set of human GAPDH was the same as mentioned before.

## β-Globin PCR amplification

Four different length fragments of the housekeeping gene β-globin were amplified to evaluate DNA quality (*Le Page et al., 2013*). The maximum amplicon size positively correlates with DNA quality. DNA samples were of good quality with at least three amplified ß-globin bands of increasing size (*Le Page et al., 2013*). PCR amplification was performed with 50 ng of tissue DNA using DNA Polymerase High Fidelity (TransTaq; TransGen Biotech Co., Ltd., Beijing, China). The primers are shown in Table 1. PCR was performed using the following conditions: primer I: initial denaturation at 94 °C for 5 min, followed by 35 cycles of denaturing at 94 °C for 30 s, annealing at 56 °C for 30 s, extension at 72 °C for 30 s, and final extension at 72 °C for 5 min; primer II: initial denaturation at 94 °C for 5 min, followed by 35 cycles of denaturing at 94 °C for 30 s, annealing at 52 °C for 30 s, extension at 72 °C for 30 s, and final extension at 72 °C for 5 min; primer III : initial denaturation at 94 °C for 5 min, followed by 35 cycles of denaturing at 94 °C for 30 s, annealing at 52 °C for 30 s, extension at 72 °C for 1 min, and final extension at 72 °C for 5 min, primer IV : initial denaturation at 94 °C for 5 min, followed by 35 cycles of denaturing at 94 °C for 30 s, annealing at 54 °C for 30 s, extension at 72 °C for 1min, and final extension at 72 °C for 5 min. PCR products were analyzed on agarose gels.

## Statistical analysis

Statistical differences were analyzed using one-way ANOVA test. The criterion for significance was set at $P$ value <0.05.

## RESULTS

### Composition of the biobank

The total number of cancer cases reached 3,686 from 2010 to 2016. 3,686 patients were consented and tissue was collected from each patient. Figure 1 showed the constituent ratio

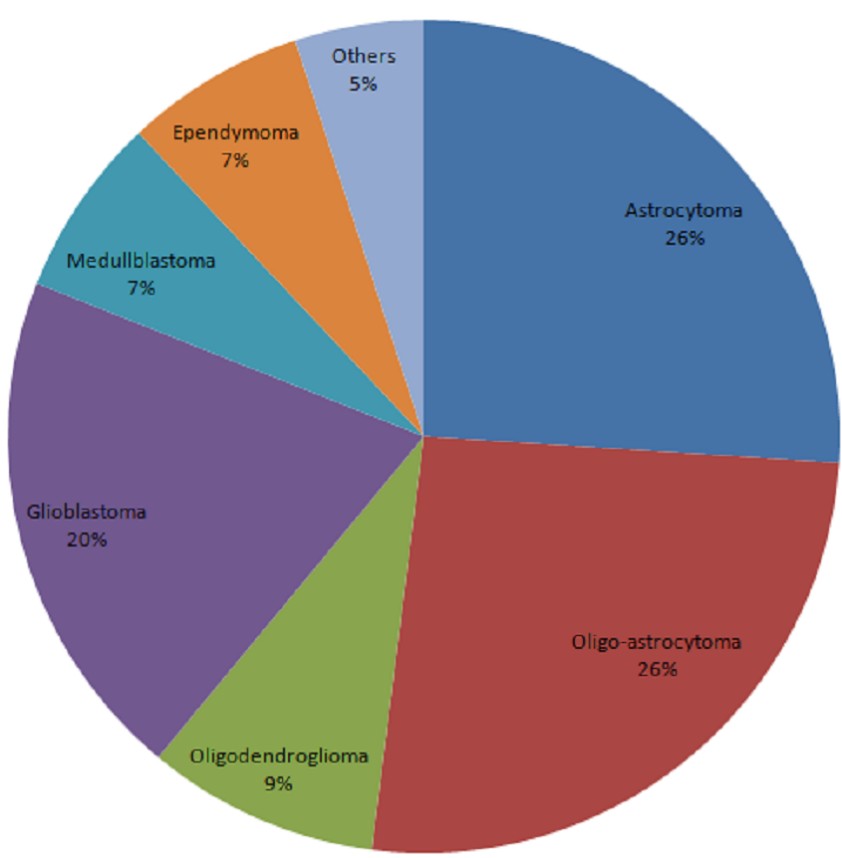

**Figure 1 Constituent ratio of glioma subtypes.**

of glioma subtypes. The vast majority subtypes of the biobank were oligo-astrocytoma, astrocytoma and glioblastoma.

## Morphology characteristics

According to the HE staining results, all tumor samples were comprised of ≥80% tumor nuclei and ≤20% necrosis (≤50% necrosis for GBM). Figure 2 shows the representative images from the 100 samples. No significant tissue decomposition was detected in all selected samples.

## RNA integrity of different storage duration

The majority of RNA samples had a A260/A280 ratio ranging between 1.8 and 2.1 which indicate high purity (Fig. 3). The RIN results of the selected 100 samples showed that 95% samples in the biobank were suitable for RT-qPCR and gene array analysis with RIN ≥6 (Fig. 4). Besides, no significant change in the RNA quality was found among different storage periods (from 2011 to 2015).

## RT -qPCR for Housekeeping genes

RT -qPCR for housekeeping genes (GAPDH, ACTB) was performed to verify the accuracy of RIN determination. Except for the 5% samples with RIN <6, of which the ct values were

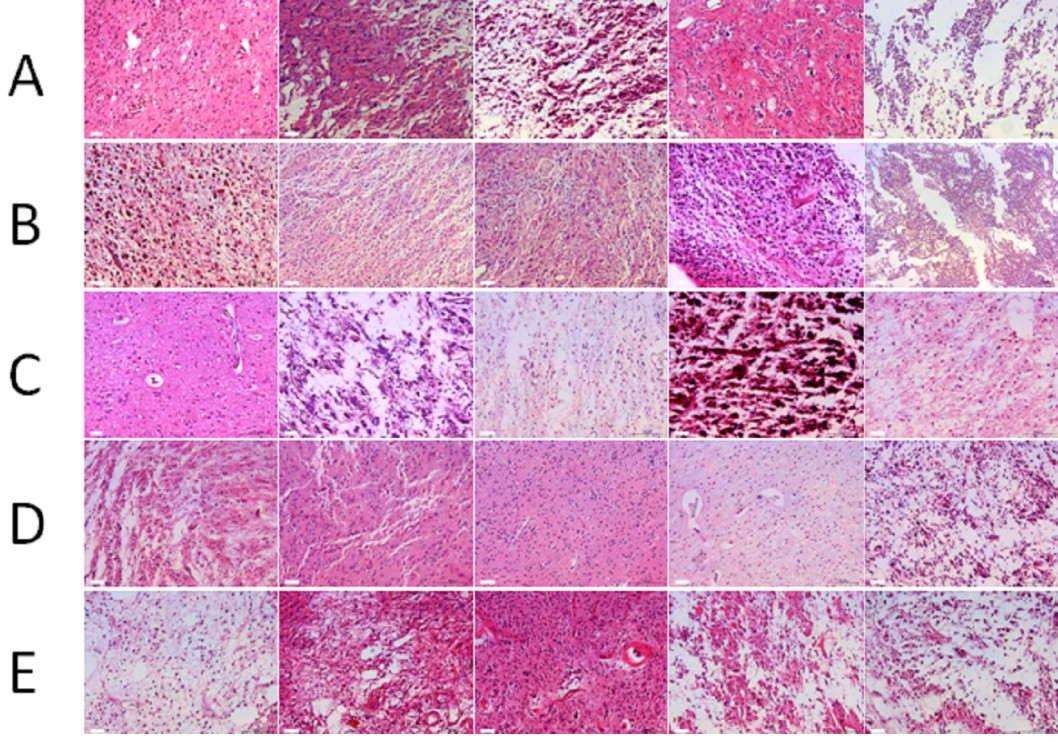

**Figure 2  Morphology of glioma tissues preserved in liquid nitrogen of different storage periods (from 2011 to 2015).** (A) Morphology of glioma tissues preserved in liquid nitrogen in 2011. (B) Morphology of glioma tissues preserved in liquid nitrogen in 2012. (C) Morphology of glioma tissues preserved in liquid nitrogen in 2013. (D) Morphology of glioma tissues preserved in liquid nitrogen in 2014. (E) Morphology of glioma tissues preserved in liquid nitrogen in 2015.

relatively higher, gene expression levels of all the other 95% samples remained stable under different storage periods (Table 2 and Fig. 5). No significant difference was found.

## DNA integrity of different storage durations

The A260/A280 and PCR for genomic housekeeping genes (GAPDH, ACTB) were performed to evaluate DNA degradation level. DNA samples had a A260/A280 ratio ranging between 1.8 and 2.0 with the exception of two samples showed ratios of 2.05 and 2.07 (Fig. 6), suggesting contamination during the phenol extraction. According to the PCR results, all the ct values were stable of different storage durations, no significant difference was found (Fig. 7). DNA quality was assessed also by PCR amplification of ß-globin gene fragments. The integrity of DNA was scored according to the number of amplified bands of increasing size from 1 to 4. According to our results, 100% of extracted DNA samples were of good quality with at least three amplified ß-globin bands of increasing size (Table 3). No significant difference was found in DNA quality of different storage durations.

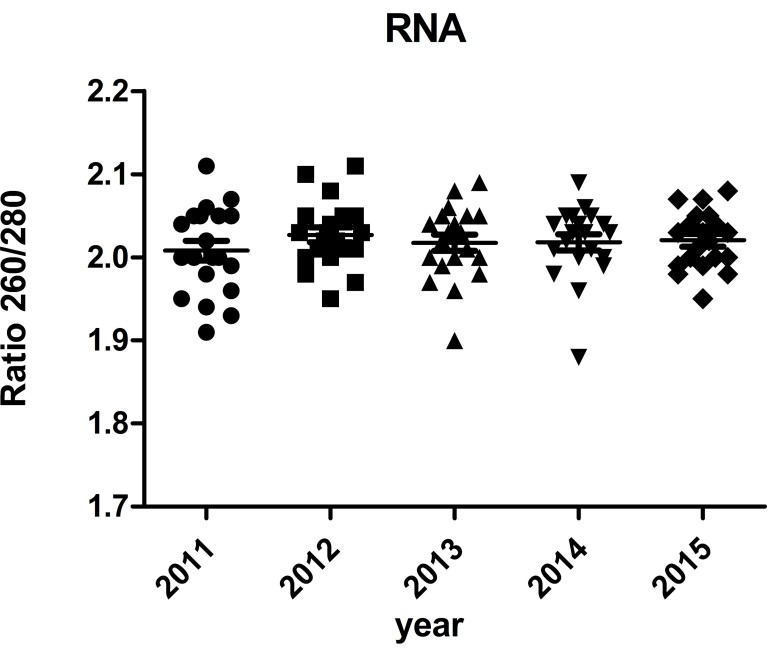

Figure 3 Ratio 260/280 of RNA samples.

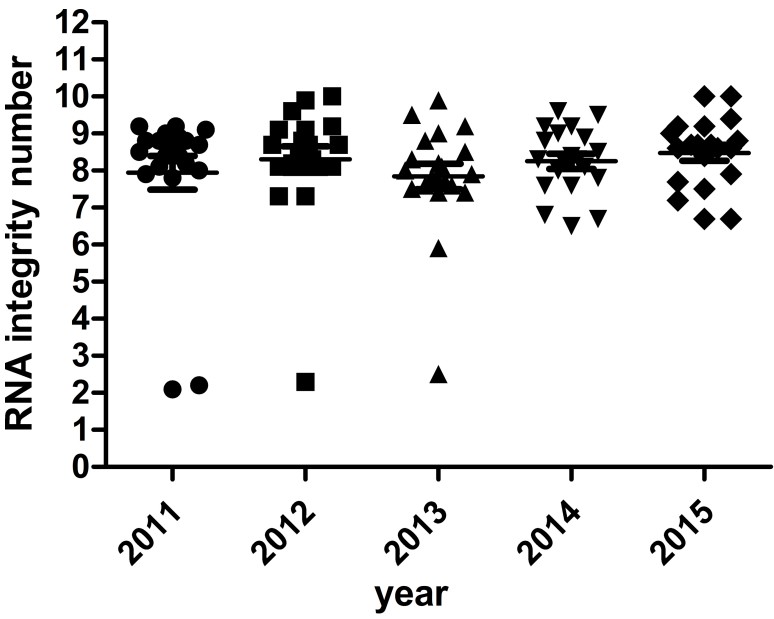

Figure 4 RNA integrity number of selected samples of different storage periods.

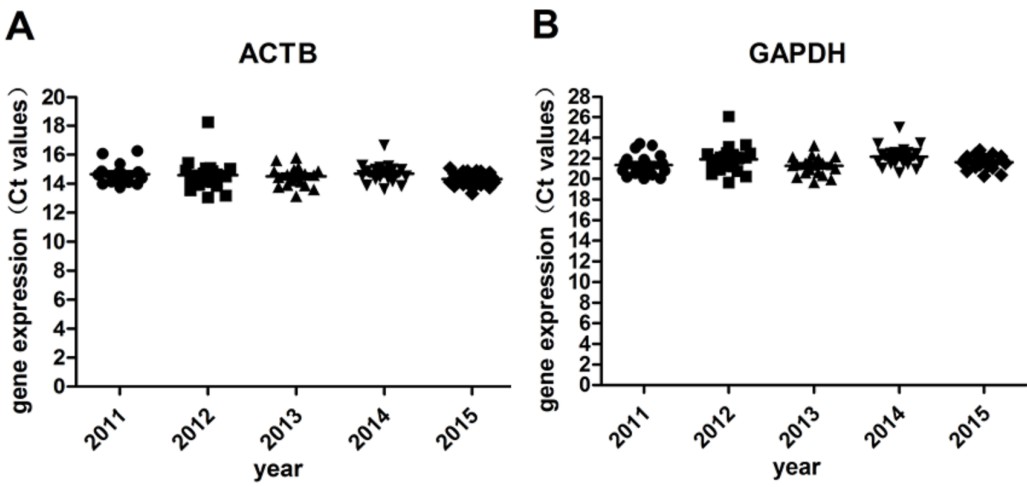

**Figure 5** **Gene expression levels under different storage durations.** (A) Gene expression levels of ACTB; (B) Gene expression levels of GAPDH.

**Table 2** **Gene expression levels under different storage durations.**

| Year | gene | Ct values (Average ± SD) of samples with | | |
|---|---|---|---|---|
| | | RIN <5 | 5 ≤ RIN < 6 | RIN ≥ 6 |
| 2011 | ACTB | 16.17 ± 0.14 | | 14.50 ± 0.41 |
| | GAPDH | 23.21 ± 0.28 | | 21.13 ± 0.85 |
| 2012 | ACTB | 18.26 ± 0.00 | | 14.39 ± 0.66 |
| | GAPDH | 26.10 ± 0.00 | | 21.69 ± 1.00 |
| 2013 | ACTB | 15.79 ± 0.00 | 15.11 ± 0 | 14.40 ± 0.58 |
| | GAPDH | 23.24 ± 0.00 | 22.10 ± 0 | 21.11 ± 0.75 |
| 2014 | ACTB | | | 14.70 ± 0.65 |
| | GAPDH | | | 22.17 ± 1.01 |
| 2015 | ACTB | | | 14.32 ± 0.47 |
| | GAPDH | | | 21.62 ± 0.66 |

## DISCUSSION

Tumor biobanks aim to collect and store sufficient number of tumor samples with high quality for basic cancer research. Clinical and basic researchers use these samples to carry out molecular biology, cell biology, genetics, transcriptomics, genomics and proteomics research in order to explore new standard of tumor classification, diagnosis, treatment and prognosis. Prolonged storage periods of tumor samples allows researchers to design studies to identify biomarkers of aggressiveness and responses to different drug treatments, which increases their value (*Hewitt, 2011*; *Olivieri et al., 2014*). Thus, establishment of tumor biobanks with high quality samples plays an important role in personal medicine and translational research.

Compared with other moleculers, the RNA molecule is less stable and more likely to degrade by RNases because its ribonucleotides contain a free hydroxyl group in the
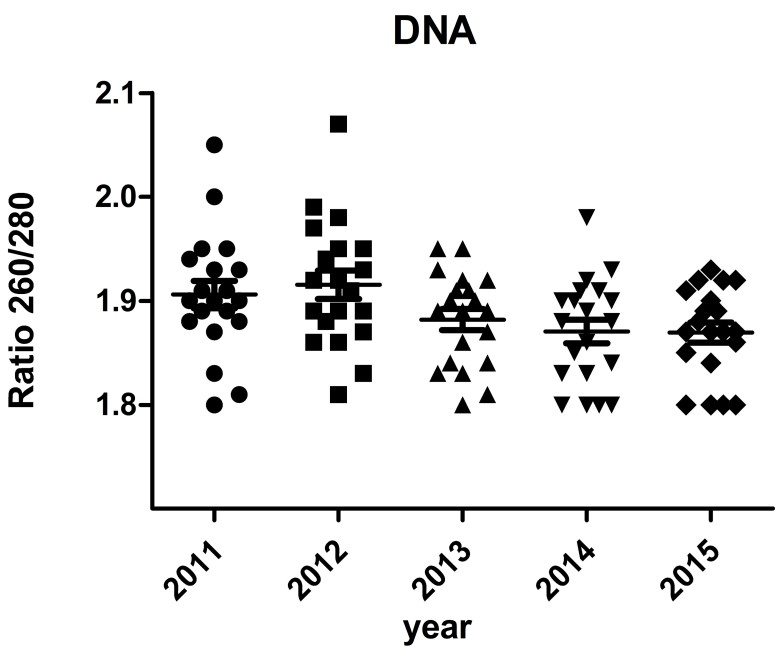

**Figure 6 Ratio 260/280 of DNA samples.**

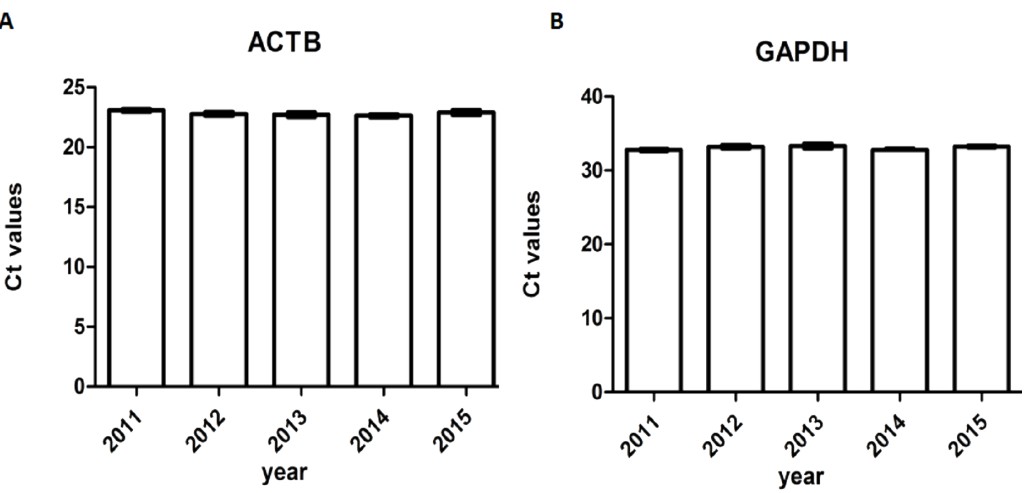

**Figure 7 Ct values of genomic housekeeping genes under different storage durations.** (A) Ct values of ACTB (B) Ct values of GAPDH.

pentose ring (*Zhang et al., 2016*). However, in our study, we didn't find significant RNA degradation in samples preserved in the liquid nitrogen tank (−196 °C) for 1–6 years (from 2011 to 2015). Except for few samples (5%) with RIN < 6, the majority of them (95%) remain high quality for further analysis with RIN ≥ 6. We verified the accuracy of RIN determination by RT-qPCR of housekeeping genes (GAPDH, ACTB). The number of cycles required to reach a detectable threshold level for fluorescence is defined as ct. Ct is

**Table 3   Number of samples with β-globin bands amplified of each year.**

| Primer | Number of samples of different storage periods | | | | |
|---|---|---|---|---|---|
| | 2011 | 2012 | 2013 | 2014 | 2015 |
| I | 20 | 20 | 20 | 20 | 20 |
| II | 20 | 20 | 20 | 20 | 20 |
| III | 20 | 20 | 20 | 20 | 20 |
| IV | 15 | 13 | 16 | 19 | 15 |

inversely correlated with the amount of template RNA (*Riemer, Keskin & Reinherz, 2012*). Thus, RNA samples with high quality have a relatively lower ct value whereas RNA samples with degradation have a relatively higher ct value of housekeeping genes. Our results of RT-qPCR were in accordance with RIN determination. All the 95% samples with RIN $\geq 6$ have stable ct values of housekeeping genes (ACTB: $14.32 \pm 0.47 \sim 14.70 \pm 0.65$; GAPDH: $21.11 \pm 0.75 \sim 22.17 \pm 1.01$). Our findings were consistent with those of *Andreasson et al. (2013)*, who assessed RIN values in endocrine tissues stored at $-80\,°C$ for approximately 30 years and found that long-term storage in $-80\,°C$ did not adversely affect the quality of the RNA extracted from the tissues. In our study, we found that glioma tissues preserved in liquid nitrogen ($-196\,°C$) could well maintain RNA quality for at least 5 years. There are also other studies showed similar results such as *Hebels et al. (2013)* detected RIN values in blood samples stored at $-80\,°C$ for 4–19 years and at $-196\,°C$ for 11–19 years and found no adversely correlation between RNA quality and storage duration. DNA and protein are more stable than RNA. QPCR reactions of genomic housekeeping genes (GAPDH, ACTB) showed no signs of DNA degradation as the ct values were stable (ACTB: $22.63 \pm 0.63 \sim 23.06 \pm 0.57$; GAPDH: $32.77 \pm 0.63 \sim 33.32 \pm 1.50$). β-Globin PCR amplification showed that 100% of extracted DNA samples were of good quality with at least three amplified ß-globin bands of increasing size.

According to our research, storage conditions of our biobank are suitable for long-term (at least five years) sample preservation with high moleculer quality. However, there are also limitations of our research. Heterogeneity has been found in various human tumors and glioblastoma (GBM) is a highly heterogeneous tumor (*Diaz-Cano, 2012*; *Furnari et al., 2015*). In our research, we only studied the tumor nuclei and necrosis proportion, but we did not study whether the pathological type of samples preserved in our biobank is consistent with the previous pathological diagnosis. We only studied gliomas and are not sure whether the other tumors have similar quality results. Moreover, our biobank established only for a short period (presently up to six years), we should further verify the quality over time to find optimal storage periods.

## CONCLUSION

Storage conditions of our biobank are suitable for long-term (at least five years) sample preservation with high molecular quality.

## ACKNOWLEDGEMENTS

The authors kindly thank staffs of the Neurosurgical department of Beijing Tiantan Hospital in helping with the collection of tissue samples.

### Funding

This study was supported by the Natural Science Foundation of China (nos. 81572474), and the Science and Technology Development Fund Project of Traditional Chinese Medicine of Beijing (JJ2015-14). The funders had no role in study design, data collection and analysis, decision to publish, or preparation of the manuscript.

### Grant Disclosures

The following grant information was disclosed by the authors:
The Natural Science Foundation of China: 81572474.
Traditional Chinese Medicine of Beijing: JJ2015-14.

### Competing Interests

The authors declare there are no competing interests.

### Author Contributions

- Fanhong Kong conceived and designed the experiments, performed the experiments, analyzed the data, contributed reagents/materials/analysis tools, prepared figures and/or tables, authored or reviewed drafts of the paper, approved the final draft.
- Wenli Zhang, Lin Qiao, Qi Li, Haowen Li, Jingli Cao, Wenyan He, Chengya Dong, Yanjiao He, Lu He, Li Liu, Weilun Fu, Lijun Liu and Zirui Li performed the experiments, contributed reagents/materials/analysis tools, authored or reviewed drafts of the paper, approved the final draft.
- Yajie Wang conceived and designed the experiments, performed the experiments, contributed reagents/materials/analysis tools, authored or reviewed drafts of the paper, approved the final draft.

### Human Ethics

The following information was supplied relating to ethical approvals (i.e., approving body and any reference numbers):

Capital Medical University affiliated Beijing Tiantan Hospital granted Ethical approval to carry out the study within its facilities (Ethical Application Ref: KY2014-021-02).

### Data Availability

The raw data has been provided as a Supplemental File.

### Supplemental Information

Supplemental information for this article can be found online at http://dx.doi.org/10.7717/peerj.4450#supplemental-information.

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
