# Peer review of "Establishment and quality evaluation of a glioma biobank in Beijing Tiantan Hospital"

_PeerJ, doi:10.7717/peerj.4450_

## Round 0.1 · original submission · Major Revisions

· Academic Editor

Major Revisions

Both reviewers have acknowledged that this is a well-designed study that evaluates the quality and suitability of tissues for analysis after cold storage. However, the article is not suitable for publication at this stage due to the limited set of parameters analysed. The authors should feel free to respond to all comments by both reviewers in addition to the clarifications provided below.

Both reviewers have highlighted an issue with the assays used in the analysis of DNA, RNA, Protein and tissue integrity obtained from the archived materials. This is particularly evident in the use of a single technique each for DNA, RNA, Protein and tissue integrity analysis, and in the use of one or at most two targets in these assays. These limitations affect the ability to verify overall quality of nucleic acids, the fragment size range and extent of degradation of RNA, the stability of post-translational modifications on proteins and antigen availability for immuno-histochemical analysis, for example.

The authors should address these deficits in a revised submission and clearly label the new experiments that have been included to strengthen the sample integrity analysis of DNA, RNA, Protein and Tissues. An extensive rewrite will also be needed, to emphasise the limitations with the current study in relation to ‘omic’ analyses, in particular, and also in relation to common QA/QC assays used by similar biobanks. The extensive rewrite should also paying particular attention to clarity, grammar and vocabulary throughout the text and as highlighted by the reviewers below.

Reviewer 1 ·

Basic reporting

The article's English language quality is fairly good but it can be improved.

Some points need to checked/clarified:
- lines 81, 116, 128: use of 100g or 100mg or tissue?
- lines 49, 182, meaning of "period of storage" unclear: Samples were collected during the period 2011-2015. The storage period refers to the time between collection and analysis - eg if the analysis was conducted in 2016, the storage period may have varied between 1 and 6 years and not "at least 5 years" as mentioned in the text.
- lines 80-82: If the frozen tissue was placed into the tube containing TRIZOL (liquid), pulverization would generate a homogenate, not a powder, and it would be more appropriate to speak of "homogenisation" rather than "pulverization".
- line 129: what does the RIPA reagent do? i.e. why was this incubation done? e.g. to solubilize proteins?

Experimental design

The basic aim of the study, ie to evaluate the quality and suitability for specific types of analysis of tissues kept in cold storage for a number of years is well defined and the study design is generally appropriate. Rightly attention is focused on the quality of RNA, a particularly unstable material, and its appropriateness for different types of analysis. Regarding the latter, only suitability for RT-PCR was tested, focusing on just 2 genes. An analogous but more serious problem exists regarding the quality assessment of the protein content of the samples, where only a single protein was evaluated.

Validity of the findings

The evaluation of RNA quality by RT-PCR for just 2 genes is far from sufficient to allow any generalisation regarding the state of RNA molecules in general and some global gene expression analyses (microarray or sequencing-based) would be necessary for such an evaluation. While the RIN numbers suggest that most of the samples would be OK for transcriptomics, the limitation represented by the absence of global analyses should be noted and discussed. Similarly, the size analysis of just one protein says very little regarding the state of the samples, making statements like "Protein degradation was not found from western blot analysis" inappropriate.

The study's conclusions need to be modifed to acknowledge the above important limitations.

Reviewer 2 ·

Basic reporting

This article describes the efforts of a glioma biobank to test the quality of samples that have been stored in liquid nitrogen for up to 5 years. The authors provided sufficient background and, for the most part, included sufficient references (exceptions are noted below), within their manuscript. The structure of the article conforms to the standard format and is a self-contained work.

Overall, the English language should be improved. In many cases the wrong word is used, there are grammatical mistakes, or articles are missing or misused. For example line 29, the word “remain,” line 45 the word “advantage”, line 72 the words “following evaluation,” or line 208 the word “pattern” were the wrong word choice and make the text difficult to understand. It was also unclear what the author’s meant by a “standard” glioma biobank (lines 22 and 46). The author’s would benefit from having a native English speaker edit their work.

Raw data has been made available; however, the consent document was in Chinese, so I was unable to evaluate its content. Details about figures and raw data are discussed in following sections.

Experimental design

This publication includes primary research that is within the Aims and Scope of the journal. The research question (whether the samples stored within the biobank maintain their quality over time) was well-defined. This manuscript provides some evidence to support the author’s assertion that their samples are of good quality and maintain that quality over time. However, issues with their assay choices (as described below) limit its usefulness to other biobanks who may want to apply their techniques or to researchers requesting their samples who want to be assured of their quality.

The methods, for the most part, were well described. Clarification on the following would improve the ability for researchers to understand the methods and replicate the findings:
• Line 62, it’s unclear what the authors mean by “the rest of the tissue samples were gathered.”
• Line 73, clarify whether the tissue samples were thawed before being placed in formalin or not.
• Line 109, clarify if the “measurement” that was performed in triplicate was the entire qPCR experiment or the same sample was run three different times in the same experiment.
• Line 134-136, the authors should specify how much protein was run from each sample on the gel (assuming it was the same amount for each sample) and should provide specifics about the product number and details (polyclonal or monoclonal) of the anti-ACTB antibody.

Validity of the findings

Composition of the Biobank:
• Clarification in line 145 if 3,686 patients were consented and tissue was collected from each patient or if 3,686 tissue aliquots have been collected from a smaller number
of consented patients.

Morphological Characteristics:
• Figure 2, the authors should indicate whether or not these are representative images from the 100 samples they looked at or they only looked at 25 H&E slides.
• In line 150, how was “cellular decomposition” evaluated or quantitated (if it was quantitated)?

RNA analysis:
• The authors should have shown or mentioned the A260/280 data to provide additional support (or as a comparison to the RIN and qPCR data) for their RNA quality results.
• Table 1 and Figure 3 are redundant.
• In Table 2, it is unclear why samples without a RIN score would have acceptable Ct values. This should be explained in Table 2 and in the text.
• In Figure 3, the authors should indicate why several RIN scores were N/A. In these cases, could the samples have been run again to get a RIN score?

DNA quality analysis:
• When evaluating DNA quality, the authors should have provided the 260/280 results as an indicator of DNA purity in their genomic DNA preps.
• Genomic DNA quality can only be measured by qPCR analysis if the primers used result in long reads, which would show that the DNA has not degraded. For example, in LePage et al (Biopreservation and Biobanking April 2013: 83-93) http://online.liebertpub.com/doi/pdfplus/10.1089/bio.2012.0044, the authors use qPCR to evaluate DNA quality looking at 4 different length fragments of the beta-globin gene. Could the authors please clarify how their qPCR evaluates DNA quality?
• Figure 5 and Table 3 are redundant.
• On lines 119-126, the authors should change RT-qPCR to just qPCR (because there is no reverse transcription step when analyzing genomic DNA) and change cDNA to genomic DNA.

Western Blot:
• It is unclear how one Western blot looking at one gene is able to assess protein quality or degradation. Could the author’s provide additional details?
• Most of the original files for the western blots appear to have increased contrast, making it difficult to evaluate if there are other bands or protein degradation on the blot.

Discussion:
The authors conclude that RNA, DNA, and proteins maintain their high quality even after being stored for up to 5 years. However, based on the analyses presented throughout the paper, I am unconvinced that their data provides the evidence tosupport this conclusion. In particular, the protein and DNA quality assays, I do not believe show that the proteins or DNA are not degraded or not. Perhaps the authors could provide more justification as to why they drew these conclusions. The authors also mention that the Ct values were “stable” (lines 190 and 200), which they conclude means that there wasn’t RNA or DNA degradation. It’s unclear what is meant by “stable” and how this relate to RNA or DNA quality.

Additional comments

No additional comments

---

## Round 0.2 · Minor Revisions

· Academic Editor

Minor Revisions

Please address, point by point, the reviewer's comments.

Reviewer 2 ·

Basic reporting

The English language in the article continues to need some work. As stated in the original review, there continue to be issues with grammar, articles, spelling, and misused words. Examples of this include (but are not limited to): “staffs” on lines 23 and 62, “the word choice of “remain” on line 29, the use of the work “normative on line 48,the verb “were” (should be “was”) on line 71, word choice “removed” on line 86, misspelling on “moleculers” on line 215, etc. The author’s would benefit from review by an English-speaking editor.

Raw data was provided, however I was unable to open the .rar file containing the Beta-globin results and therefore could not evaluate that data.

Experimental design

The majority of the concerns with experimental design were addressed in the updated manuscript except for adding information about the antibodies used for the Western Blot.

The author’s added Beta-globin analysis to look at DNA quality, but did not provide rationale for this type of experiment or the reference citing the LePage article. Both would be needed additions.

Validity of the findings

The author’s addressed the majority of the concerns from the initial review. The addition of the A260/A280 data figure was appropriate and this reviewer appreciated the author's re-running the samples with N/A for RIN scores from the original manuscript. The addition of the Beta-globin analysis for DNA quality should be shown as a figure or be provided in an accessible format for evaluation and as supplemental information.

The Western Blot and protein quality data does not assess protein quality. Although the authors add text in the discussion saying that this experiment doesn’t mean that all proteins are of good quality, this reviewer would suggest that the protein analysis data be removed from the manuscript altogether. It does not add anything of value to the article.

---

## Round 0.3 · accepted · Accept

· Academic Editor

Accept

The authors addressed all the comments and the article is eligible for publication.